# Eleven internal fixations for young vertical femoral neck fractures: A protocol for systematic review and network meta-analysis

Weiwei Shen[1], Yun Xue[1]*, Jie Shi[1], Xiaowen Deng[1], Zhongshu Pu[2], Qiuming Gao[1]

**1** Orthopedics Center of PLA, The 940th Hospital of Joint Logistics Support Force Army of PLA, Lanzhou, Gansu Province, China, **2** Department of Infection Control, The 940th Hospital of Joint Logistics Support Force Army of PLA, Lanzhou, Gansu Province, China

\* xueyuncloud@163.com

## Abstract

### Background and objectives

Vertical femoral neck fractures (VFNFs) in young patients lead to significant biomechanical instability. Multitudinous internal fixation devices have been developed and utilized in clinical interventions. However, there has yet to be a consensus expert opinion regarding the optimal internal fixation configurations. This study aims to conduct a network meta-analysis to evaluate the safety and efficacy of all currently recognized internal fixation procedures for the treatment of VFNFs in young individuals.

### Methods

Comprehensive literature searches will be performed in China National Knowledge Infrastructure, the Cochrane Library, PubMed, Web of Science, Embase, the Wanfang Database, and the Chinese Biomedical Literature Database, covering the entire database history up until May 21, 2024. Individual papers will be evaluated for possible bias using RoB 2.0, the most recent version of the randomized trial Cochrane risk-of-bias approach. Pairwise meta-analysis and network meta-analysis (NMA) will be conducted for data analysis using STATA 15.0 and R 4.1.3. Inconsistency tests, subgroup analyses, sensitivity analyses, and assessments of publication bias will also be performed.

### Conclusion

The study will provide evidence-based recommendations for the optimal internal fixation methods in treating young patients with VFNFs.

### Trial registration

INPLASY202460017.

**Data Availability Statement:** No datasets were generated or analysed during the current study. All relevant data from this study will be made available upon study completion.

**Funding:** The work is supported by the Scientific and Technological Planning Projects of Gansu Province (No. 21JR7RA016) and Key Clinical Specialties of PLA. The funders had no role in study design, data collection and analysis, decision to publish, or preparation of the manuscript.

**Competing interests:** The authors have declared that no competing interests exist.

## Introduction

Femoral neck fractures (FNFs) are the predominant type of fracture encountered in clinical practice, constituting 60% of all hip fractures [1]. The worldwide incidence of hip fractures, estimated at 1.3 million annually in 1990, is projected to increase to 2.6 million by 2025 and reach 4.5 million by 2050 [2]. Vertical FNFs (VFNFs) in young patients primarily occur due to high-force incidents, including automobile accidents and forceful collisions. These fractures lead to increased biomechanical instability [3]. To provide a successful clinical outcome for these fractures, effective internal fixation and anatomic reduction are commonly acknowledged [4]. Unfortunately, there is a significant prevalence of postoperative problems after performing internal fixation on vertical fractures in young patients. Approximately 45% of young patients with FNFs experience problems, including failed fixation, nonunion, malunion, and osteonecrosis. Out of these patients, 32% require significant reconstructive surgery [5].

In recent times, multitudinous internal fixation devices have been developed and used in clinical practice, such as proximal femoral nail anti-rotation (PFNA), cannulated compression screw (CCS), dynamic hip screw (DHS), medial buttress plate (MBP), and femoral neck system (FNS). However, there is no consensus expert opinion on the most efficient internal fixation configurations. Many biomechanical investigations and clinical trials have been performed to find the most effective arrangements of internal fixations for VFNFs. Several researches have revealed that the inverted triangle arrangement of three cannulated screws exhibits greater strength compared to the configuration with triangle screws [6, 7]. Meanwhile, some other studies have demonstrated that DHS is more effective compared to cancellous screw constructs, as DHS could resist more shear forces in vitro biomechanical tests [8, 9]. The FAITH-2 research (Fixation Using Alternative Implants for the Treatment of Hip Fractures), which involved multiple countries and centres, conducted a randomized controlled experiment in young femoral neck fracture patients [10]. The research showed insignificant differences in the outcomes between cannulated screw constructs and fixed-angle devices up till now.

Multiple meta-analyses and systematic reviews have documented the comparative analysis of various forms of internal fixations. Xia et al. [11] reported a higher incidence rate of avascular necrosis of the femoral head with SHS, whereas CCS showed a higher rate of implant loosening. Zhou et al. [12] performed a meta-analysis comprising 21 trials and 1,347 patients. Their findings indicate that FNS is better than CCS in terms of both effectiveness and safety for the internal fixation of FNFs. However, the traditional meta-analysis has not been able to compare all treatment strategies for young VFNFs thoroughly. The emerging network meta-analysis (NMA) has gained a significant acceptance in medical research and is recognized as yielding high-quality evidence, because of the ability to simultaneously compare multiple interventions directly and indirectly [13, 14]. Zhang et al. [15] conducted the first NMA to compare and rank four surgical procedures for the elderly displaced FNFs. This NMA focused on the difference between hemiarthroplasty, internal fixation, and total hip arthroplasty. However, it didn't detail the differences between various internal fixations. This is a significant inadequacy, as existing evidence has demonstrated that young individuals with VFNFs should have internal fixation for treatment [16]. So far, numerous internal fixation devices have been developed and utilized in the interventions of VFNFs, including CCS, DHS, PFNA, and FNS [17, 18]. Meanwhile, many modified techniques and configurations of CCS have been clinically proven effective, such as adding a fourth Pauwel screw [19], and a biplane double-supported screw fixation method [20]. Therefore, there is an urgent need to update this NMA and solve the deficiency.

This study aims to conduct the NMA of published randomized controlled trials (RCTs) in order to assess the effectiveness of eleven different internal fixations for young patients with

VFNFs. The fixation techniques under evaluation include: 1) three parallel screws with inverted triangular (ITR) or 2) triangular (TRI) configurations, as well as configurations involving 3) inverted parallel screws with a Pauwel screw, arranged in an "alpha" configuration (ALP); and 4) with a buttress plate (BUT) strengthening the calcar (BUT); 5); biplane double-supported screw fixation (F-technique); 6) four parallel screws arranged in a "rhomboid" configuration (RHO); 7) dynamic hip screw fixation without anti-rotation screws (DHS); 8) dynamic hip screw fixation with anti-rotation screws (DHS+); 9) femoral neck system (FNS); 10) cephalomedullary nails (CMN); 11) proximal femoral plates (PFP). It is anticipated that this novel NMA will furnish empirical medical data for the management of juvenile VFNFs using internal fixations, hence enhancing support for orthopaedic practitioners.

## Methods

Both this systematic review and the NMA protocol have followed the guidelines set forth in the Preferred Reporting Items for Systematic Reviews and Meta-Analyses protocols (PRISMA-P) [21]. The checklist of PRISMA-P is available in S1 File on the website. Our protocol is now officially registered on the INPLASY database under the reference number INPLASY202460017.

### Eligibility criteria

The criteria for inclusion or exclusion of studies are formulated based on the PICOS framework, which encompasses factors related to the patient population, comparator, outcomes, intervention, and study design.

**Participants (P).**   All young patients who have been diagnosed with VFNFs using X-plain film or CT scan will be included in our systematic review, independent of their country of origin, race, or gender. Patients with pathological VFNFs or those who are elderly and have a femoral neck fracture (age $\geq$ 60 years) will be excluded.

**Interventions (I).**   There will be more than or equal to one form of internal fixations for juvenile VFNFs that will be included as part of the experimental group. These include ITR, TRI, ALP, RHO, BUT, F-technique, DHS, DHS+, FNS, CMN, and PFP fixes. Furthermore, operations including hemiarthroplasty or complete hip arthroplasty, free vascularized fibula grafting, and valgus osteotomy will not be permitted, regardless of whether cement or noncement is used. Regarding reduction techniques (close versus open) or surgical approaches (later versus anterior versus anterolateral), there will be no limitations placed on either.

**Comparator (C).**   Patients who received another form of internal fixation treatment will be included in the contrast group.

**Outcomes (O).**   The primary outcome is defined as revision surgery regardless of study periods or any reason. Secondary measure outcomes that will be included are duration of surgery, intraoperative blood loss, osteosynthesis failure or loosening, Harris Hip Score (HHS), length of stay in the hospital, deep vein thrombosis, mortality, and postoperative complications, including surgical nonunion, site infections, and avascular necrosis of the femoral head (ANFH).

**Study design (S).**   This analysis will just use RCTs. Case-control studies, opinion papers, editorials, reviews, case reports, and cohort studies will not be considered.

### Data repositories and search plan

A thorough search will be carried out in the following databases: China National Knowledge Infrastructure, PubMed, Embase, the Wanfang Database, the Chinese Biomedical Literature Database, Web of Science, and the Cochrane Library. The search will encompass all available

data from the inception of each database up until May 21, 2024. The search for studies will be limited to publications in either Chinese or English. The literature search will integrate both free text words and MeSH terms to determine related trials. The details of the strategy of PubMed search can be found in S2 File. Search techniques specific to each database will be modified as necessary. A manual review of eligible studies and related systematic reviews will be conducted in order to identify any missed research or additional data.

## Study selection

The Zotero 6.0 software will be used to import all of the retrieved research, and any duplicates will be eliminated thereafter. The abstracts and titles of all the identified studies will undergo independent scrutiny by two reviewers. After the preliminary exclusion of irrelevant publications, all potentially relevant studies' full texts will be downloaded, and reviewed for further independent assessment. Any conflicts will be handled by collaborative discussion within the team or by seeking input from a third reviewer. Finally, the selected papers will be presented in a flow diagram (Fig 1) following the guidelines of PRISMA-P.

## Data acquisition

Two reviewers will independently use a pre-established, standardized data abstraction sheet to extract data from each of the listed studies. Information to be extracted includes:

1. Basic information: title, first authors, the publication year, country, and language.

2. Trial characteristics: study design, follow-up period, random method, blinding method and allocation concealment.

3. Participants: sample size of different groups, diagnostic methods, mean age, gender, Pauwels classification, fracture displace degree.

4. Interventions: internal fixation system, fracture reduction method (open vs close), and surgery times.

5. Outcomes: all specified outcomes, number of participants with complete follow-up and document reasons for loss to follow-up.

## Evaluation of bias risk

The bias risk in the included studies will be independently assessed by two reviewers using the updated Cochrane risk-of-bias tool for randomized trials (RoB 2.0) [22]. This tool analyzes bias in five different domains: the randomization procedure; outcome measurement; missing outcome data; deviations from intended interventions; and the selection of reported outcomes, each of which is classified as low risk, moderate concerns, or high risk. According to this assessment, each investigation will be classified as having a moderate, high, or low bias risk. Any disagreements will be resolved by a third reviewer or through consensus.

## Statistical analysis

**1. Pairwise meta-analysis.** Direct comparisons with R 4.1.3 will be conducted through pairwise meta-analysis employing the meta package [23]. Continuous variables will be assessed using the mean difference (MD) and 95% confidence intervals (CIs), while dichotomous variables will be analyzed with the risk ratio (RR) and corresponding 95% CI. Statistical heterogeneity among studies will be evaluated using the $I^2$ statistic and 95% prediction intervals [24,

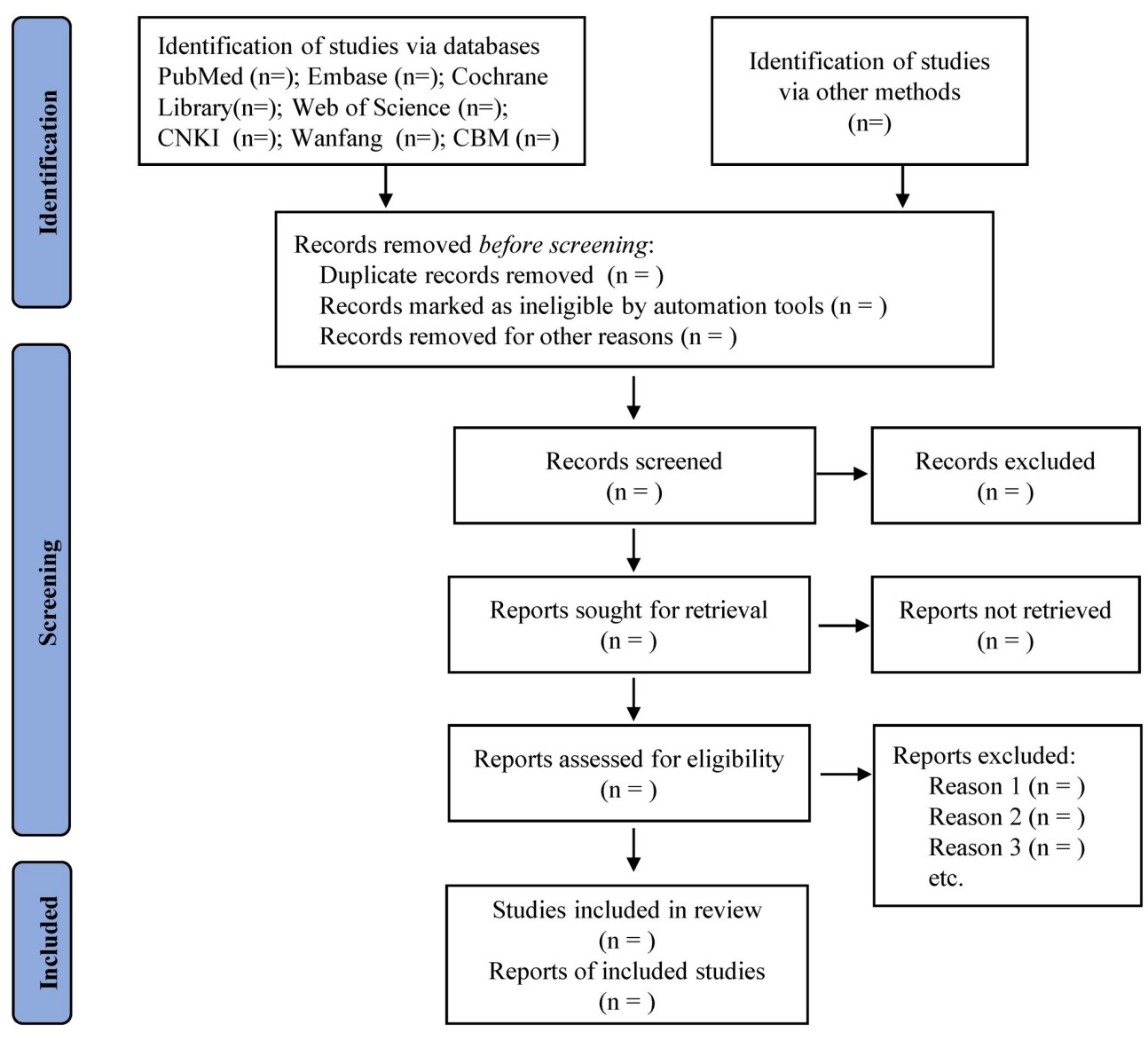

**Fig 1. Flow chart of study selection.**

25]. Due to expected heterogeneity among the studies, the random-effects model will be applied as it provides a more conservative evaluation of effect sizes.

**2. Network meta-analysis.** We will perform the NMA with R 4.1.3 software (ggplot2 package and gemtc package) on the Bayesian framework [26]. The network meta-analysis will also use random-effects models, which are considered the most conservative method for addressing heterogeneity across studies. The Markov chain Monte Carlo simulations will be built in this Bayesian NMA, with a burn-in of 10,000 iterations, sampling of 50,000 iterations and thinning interval of 10. In order to analyze the model's convergence, the Gelman-Rubin-Brooks diagnostic plots and the potential scale reduction factor will be used [27]. All potential pairwise comparisons, including mixed and indirect comparisons, will be used to summarize the NMA findings. The local inconsistency of NMA will be calculated for closed loops of interventions using the node-splitting approach [28]. To rank the interventions based on optimal

probability, the study will generate the Surface Under the Cumulative Ranking (SUCRA) curve [29]. The SUCRA value is a measure that goes from 0 to 100%. A higher value, closer to 100%, implies that a treatment has a greater therapeutic efficacy compared to other therapies investigated for the same result.

## Sensitivity and subgroup analyses

Subgroup analysis will be performed to identify sources of heterogeneity, focusing on factors such as the extent of fracture displacement and methods of reduction, among other relevant factors. An analysis of sensitivity will also be performed by systematically excluding each study to check if the findings possess stability. If there is a significant alteration in the variability or the meta-analysis result, the excluded study will be regarded as a contributing factor to the heterogeneity.

## Quality of evidence

The quality of evidence for each result in the NMA will be examined using the Grading of Recommendations, Assessment, Development, and Evaluation (GRADE) analysis [30]. Five factors contributing to potential downgrading will be evaluated: risk of bias, indirectness, publication bias, inconsistency, and imprecision. There will be 4 levels of evidence quality for each relevant outcome: extremely low, low, moderate, and high.

## Publication bias

If there are more than ten studies, publication bias will be accessed by examining comparison-adjusted funnel plots using STATA 15.0 software [31]. These funnel plots serve as a representative tool for identifying potential publication bias in systematic reviews. The publication bias will also be evaluated by employing Egger's test [32]. Asymmetry in the funnel plots suggests the existence of such bias.

## Discussion

FNFs in young patients mostly result in VFNFs, leading to greater biomechanical instability [18]. Reduction and internal fixation are widely employed as therapies in clinical practice [33]. Unfortunately, the poor outcomes and high complication rates following the internal fixation for VFNFs in young patients are substantial. The most appropriate internal fixation configurations remain controversial.

Multiple conventional pairwise meta-analyses have examined the relative effectiveness of internal fixation for VFNFs [11, 12]. However, all these only compared the therapeutic effectiveness difference between the two or three internal fixation methods. The relative effectiveness of various methods of internal fixation in young patients with VFNFs has not yet been determined. To our knowledge, this review is the first Bayesian meta-analysis aimed at assessing and ranking the most effective internal fixation methods for young VFNFs. The results of this study will enhance understanding of the benefits associated with each specific internal fixation method and offer evidence-based medical guidance for selecting appropriate procedures to treat young patients with VFNFs.

Significantly, this study also has limitations. Complete elimination of heterogeneity is challenging due to variations in fracture displacement, reduction techniques, surgical approaches, and follow-up duration. Internal fixation has demonstrated promise for treating VFNFs in young patients; however, robust multicenter RCTs are necessary to validate these findings.

## Supporting information

**S1 File. PRISMA-P (Preferred Reporting Items for Systematic Review and Meta-Analysis Protocols) 2015 checklist: Recommended items to address in a systematic review protocol.** (DOC)

**S2 File. Search strategy for PubMed.** (DOCX)

## Author Contributions

**Conceptualization:** Weiwei Shen, Zhongshu Pu, Qiuming Gao.

**Data curation:** Yun Xue, Jie Shi.

**Formal analysis:** Weiwei Shen, Zhongshu Pu.

**Investigation:** Weiwei Shen, Xiaowen Deng.

**Methodology:** Weiwei Shen, Zhongshu Pu.

**Project administration:** Weiwei Shen, Qiuming Gao.

**Resources:** Weiwei Shen, Jie Shi, Xiaowen Deng.

**Software:** Weiwei Shen, Yun Xue.

**Supervision:** Weiwei Shen, Qiuming Gao.

**Validation:** Yun Xue, Jie Shi.

**Visualization:** Qiuming Gao.

**Writing – original draft:** Weiwei Shen, Zhongshu Pu.

**Writing – review & editing:** Weiwei Shen, Qiuming Gao.

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
