## [Decision Letter · Decision Letter 0]

27 Aug 2024

PONE-D-24-24072Eleven Internal Fixations for young vertical femoral neck fractures: a protocol for systematic review and network meta-analysisPLOS ONE

Dear Dr. Shen,

Thank you for submitting your manuscript to PLOS ONE. After careful consideration, we feel that it has merit but does not fully meet PLOS ONE’s publication criteria as it currently stands. Therefore, we invite you to submit a revised version of the manuscript that addresses the points raised during the review process.

**ACADEMIC EDITOR: **According to the reviewer suggestions the paper requires some changes. Please read (at the end of the message) them carefully and include into the final version of the paper

We look forward to receiving your revised manuscript.

Kind regards,

Pawel Klosowski, D.Sc.

Academic Editor

PLOS ONE

Journal Requirements:

- Eleven Internal Fixations for young vertical femoral neck fractures: a protocol for systematic review and network meta-analysis (doi:10.37766/inplasy2024.6.0017)

In your revision ensure you cite all your sources (including your own works), and quote or rephrase any duplicated text outside the methods section. Further consideration is dependent on these concerns being addressed.

"The work is supported by the Scientific and Technological Planning Projects of Gansu Province (No. 21JR7RA016) and Key Clinical Specialties of PLA."

5. Please respond by return e-mail with an updated version of your manuscript to amend either the abstract on the online submission form or the abstract in the manuscript so that they are identical. We can make any changes on your behalf.

Reviewers' comments:

Reviewer's Responses to Questions

**Comments to the Author**

1. Does the manuscript provide a valid rationale for the proposed study, with clearly identified and justified research questions?

Reviewer #1: Yes

2. Is the protocol technically sound and planned in a manner that will lead to a meaningful outcome and allow testing the stated hypotheses?

Reviewer #1: Yes

3. Is the methodology feasible and described in sufficient detail to allow the work to be replicable?

Reviewer #1: No

4. Have the authors described where all data underlying the findings will be made available when the study is complete?

Reviewer #1: No

5. Is the manuscript presented in an intelligible fashion and written in standard English?

Reviewer #1: No

6. Review Comments to the Author

You may also provide optional suggestions and comments to authors that they might find helpful in planning their study.

Reviewer #1: I would first like to thank the Editor for inviting me to review this interesting manuscript for this prestigious journal.

My views on the requested manuscript (PONE-D-24-24072) are as follows.

# Overall opinion

- The purpose of the study is well described and convinces the reader of the need for it, and I believe this is a protocol that should be positively considered for publication.

- Once the following questions have been addressed, I would have no difficulty agreeing to publication.

# Questions or requests for clarification

1. It is recommended that risk ratios be used for calculating effect sizes in binary data, as they can be misleading to the majority of readers. Furthermore, it is appropriate to convert and report results calculated with odds ratios to risk ratios. Should authors have a specific reason for using ORs, they are asked to provide a detailed explanation in the manuscript. Otherwise, the use of RRs is to be preferred.

2. The choice between a random effects model and a fixed effects model, based on the I squared statistic, may introduce statistical bias. In the event that heterogeneity is suspected, it is recommended that the random-effects model be utilized as a default, with the fixed-effects model presented as supplementary material. This approach provides a more conservative view of the study.

3. The I squared statistic is not an absolute measure of heterogeneity. It would be prudent to include the measurement and reporting of prediction intervals in the protocol. The following recently published references may prove useful for researchers in this regard.

Suggested ref. Borenstein M. Avoiding common mistakes in meta-analysis: Understanding the distinct roles of Q, I-squared, tau-squared, and the prediction interval in reporting heterogeneity. Res Synth Methods. 2024 Mar;15(2):354-368. doi: 10.1002/jrsm.1678

4. It would be beneficial to provide an explanation, even in schematic form, of the planned categorization of primary and secondary outcomes at the protocol stage. It is essential that researchers have a clear pre-definition of the data that will be primarily reflected in the conclusions.

5. As the study employs a Bayesian statistical approach to NMA, it would be beneficial to provide a priori information regarding the number of iterations, burn-in period, and data extraction intervals for the MCMC model.

7. PLOS authors have the option to publish the peer review history of their article (what does this mean?). If published, this will include your full peer review and any attached files.

Reviewer #1: No

---

## [Author Response · Author response to Decision Letter 0]

7 Sep 2024

Response to reviews

Title: Eleven Internal Fixations for young vertical femoral neck fractures: a protocol for systematic review and network meta-analysis

Dear editor and reviewers:

Thank you for your letter and the reviewers’ comments concerning our manuscript entitled “Eleven Internal Fixations for young vertical femoral neck fractures: a protocol for systematic review and network meta-analysis” (PONE-D-24-24072). All these comments are valuable and very helpful. We have read through comments carefully and have made corrections. Based on the instructions provided in your letter, we uploaded the file of the revised manuscript. We hope the revised manuscript has now met the publication standard of your journal.

We highlighted all the revision in red. Our point-to-point responses to the reviewer's comments are presented following.

Academic editor (original comments are in blue color)

Comment 1: Please ensure that your manuscript meets PLOS ONE's style requirements, including those for file naming. The PLOS ONE style templates can be found at https://journals.plos.org/plosone/s/file?id=wjVg/PLOSOne_formatting_sample_main_body.pdf and https://journals.plos.org/plosone/s/file?id=ba62/PLOSOne_formatting_sample_title_authors_affiliations.pdf

Response: We appreciate for your warm work earnestly. On the basis of PLOS ONE's style requirements, our manuscript had been carefully checked and corrected accordingly. Revised portion are underlined in red.

Comment 2: We noticed you have some minor occurrence of overlapping text with the following previous publication(s), which needs to be addressed: 

- Eleven Internal Fixations for young vertical femoral neck fractures: a protocol for systematic review and network meta-analysis (doi:10.37766/inplasy2024.6.0017)

In your revision ensure you cite all your sources (including your own works), and quote or rephrase any duplicated text outside the methods section. Further consideration is dependent on these concerns being addressed.

Response: We fully agree with this requirement. The above-mentioned previous publication is our meta-analysis register document. This meta-analysis was registered on INPLASY, so there was some overlapping text between them to a certain extent. In fact, all the authors are well aware that copying extracts from previous publications, especially outside the methods section, word-for-word is unacceptable. The manuscript had been cross-checked by iThenticate for reviewing the text similarity. Any duplicated text had been quoted or rephrased correspondingly.

Comment 3: Thank you for stating the following financial disclosure: 

"The work is supported by the Scientific and Technological Planning Projects of Gansu Province (No. 21JR7RA016) and Key Clinical Specialties of PLA."

If this statement is not correct you must amend it as needed. Please include this amended Role of Funder statement in your cover letter; we will change the online submission form on your behalf.

Response: The funders had no role in the study. “The funders had no role in study design, data collection and analysis, decision to publish, or preparation of the manuscript.” has been stated in the revised manuscript.

Comment 4: When completing the data availability statement of the submission form, you indicated that you will make your data available on acceptance. We strongly recommend all authors decide on a data sharing plan before acceptance, as the process can be lengthy and hold up publication timelines. Please note that, though access restrictions are acceptable now, your entire data will need to be made freely accessible if your manuscript is accepted for publication. This policy applies to all data except where public deposition would breach compliance with the protocol approved by your research ethics board. If you are unable to adhere to our open data policy, please kindly revise your statement to explain your reasoning and we will seek the editor's input on an exemption. Please be assured that, once you have provided your new statement, the assessment of your exemption will not hold up the peer review process.

Response: We are all deeply appreciative of your recommendation. Taking the publication timelines problem into consideration, we have revised our data sharing plan. All relevant data from this study will be made available upon study completion. We apologize for the confusion generated by our previous data availability statement.

Comment 5: Please respond by return e-mail with an updated version of your manuscript to amend either the abstract on the online submission form or the abstract in the manuscript so that they are identical. We can make any changes on your behalf.

Response: We all appreciate for your kindness and selfless assistance. We have updated our manuscript by e-mail so that the abstract is the same on the online and the manuscript.

Comment 6: Please review your reference list to ensure that it is complete and correct. If you have cited papers that have been retracted, please include the rationale for doing so in the manuscript text, or remove these references and replace them with relevant current references. Any changes to the reference list should be mentioned in the rebuttal letter that accompanies your revised manuscript. If you need to cite a retracted article, indicate the article’s retracted status in the References list and also include a citation and full reference for the retraction notice.

Response: According to the PLOS ONE's style requirements, all reference list in our manuscript has been checked carefully and revised accordingly to ensure that it is complete and correct. No retracted papers were cited. No changes to the reference list has been made.

Reviewer's Responses to Questions (Questions are in blue color)

Question 1: Does the manuscript provide a valid rationale for the proposed study, with clearly identified and justified research questions?

Reviewer: Yes

Response: Thank you for your good comment. Our manuscript provide a valid rationale for the proposed study. Vertical femoral neck fractures (VFNFs) lead to significant biomechanical instability. However, there has yet to be a consensus expert opinion regarding the optimal internal fixation configurations. This study conducted a network meta-analysis to evaluate the safety and efficacy of all available internal fixation procedures for VFNFs.

Question 2: Is the protocol technically sound and planned in a manner that will lead to a meaningful outcome and allow testing the stated hypotheses? 

Reviewer: Yes

Response: This protocol has been reported following the PRISRM-P guideline. The criteria for inclusion studies are formulated based on the PICOS framework. Meanwhile, both conventional pairwise meta-analyses and network meta-analysis will be conducted in this meta-analysis for the treatment of VFNFs.

Question 3: Is the methodology feasible and described in sufficient detail to allow the work to be replicable?

Reviewer: No

Response: Thank you for this valuable feedback. Despite we tried our best to describe methods in sufficient detail, there would be some shortcomings, even mistakes inevitably in this protocol. We have carefully addressed all the reviewer’s concerns. All added information and correction highlighted in red have been supplied accordingly in the revised manuscript.

Question 4: Have the authors described where all data underlying the findings will be made available when the study is complete?

Reviewer: No

Response: According the PLOS Data policy, all relevant data from this study will be made available upon study completion.

Question 5: Is the manuscript presented in an intelligible fashion and written in standard English? 

Reviewer: No

Response：Thank you for pointing this out! With the help of TopEdit (www.topeditsci.com) for linguistic assistance, we carefully read the full text and corrected some language and writing problems, and marked the major changes in red. We really hope that the language level have been substantially improved.

Question 6: PLOS authors have the option to publish the peer review history of their article (what does this mean?). If published, this will include your full peer review and any attached files. 

Do you want your identity to be public for this peer review? For information about this choice, including consent withdrawal, please see our Privacy Policy.

Reviewer: No

Respond: We sincerely appreciate the valuable suggestion. After carefully reading and thoroughly understanding the PLOS Privacy Policy, we decide to publish the peer review history when an article is published.

Review Comments to the Author (original comments are in blue color)

Comment 1: It is recommended that risk ratios be used for calculating effect sizes in binary data, as they can be misleading to the majority of readers. Furthermore, it is appropriate to convert and report results calculated with odds ratios to risk ratios. Should authors have a specific reason for using ORs, they are asked to provide a detailed explanation in the manuscript. Otherwise, the use of RRs is to be preferred.

Response: We sincerely thank the reviewer for careful reading. As is well known, it is preferable to avoid reporting odds ratios in RCTs to avoid misinterpretations. So we have replaced the “ORs” with “RRs” in the reversed manuscript.

Comment 2: The choice between a random effects model and a fixed effects model, based on the I squared statistic, may introduce statistical bias. In the event that heterogeneity is suspected, it is recommended that the random-effects model be utilized as a default, with the fixed-effects model presented as supplementary material. This approach provides a more conservative view of the study.

Response: We thank the reviewer for pointing out this issue. Due to expected heterogeneity among the studies, the random effects model provides a more conservative evaluation of effect sizes. We have re-written this part according to the reviewer’s suggestion in the revised manuscript.

Comment 3: The I squared statistic is not an absolute measure of heterogeneity. It would be prudent to include the measurement and reporting of prediction intervals in the protocol. The following recently published references may prove useful for researchers in this regard. 

Suggested ref. Borenstein M. Avoiding common mistakes in meta-analysis: Understanding the distinct roles of Q, I-squared, tau-squared, and the prediction interval in reporting heterogeneity. Res Synth Methods. 2024 Mar;15(2):354-368. doi: 10.1002/jrsm.1678

Respond: We sincerely appreciate the valuable comment. We read the articles recommended by the reviewer and found a lot to learn from them. Although the vast majority of meta-analyses in medicine use I squared statistic as the primary index of heterogeneity, it is a serious mistake. The I squared index is a proportion, not an absolute amount. The prediction interval may be particularly imprecise in a meta-analysis with few studies, resulting from an imprecise estimation of the summary effect size and heterogeneity parameter. We have included prediction intervals approach in the revised manuscript, and added the suggested reference in the Reference list.

Comment 4: It would be beneficial to provide an explanation, even in schematic form, of the planned categorization of primary and secondary outcomes at the protocol stage. It is essential that researchers have a clear pre-definition of the data that will be primarily reflected in the conclusions. 

Respond: We agree with this suggestion. An explanation of the primary outcome (revision surgery) has been provided in the revised manuscript. We appreciate for your warm work earnestly.

Comment 5: As the study employs a Bayesian statistical approach to NMA, it would be beneficial to provide a priori information regarding the number of iterations, burn-in period, and data extraction intervals for the MCMC model.

Respond: We thank the reviewer for point out this issue. We indeed should have provided a priori information of MCMC model when Bayesian statistical approach is applied to NMA. The number of iterations, burn-in period, and data extraction intervals for the MCMC model is provided in the reversed manuscript.

Thank you very much for the invaluable comments and suggestions you provided on my manuscript. We have carefully considered each of your points and have responded accordingly. Should you have any further questions or suggestions regarding my paper, please feel free to contact us.

Once again, I truly appreciate your valuable time and constructive feedback.

Yours sincerely,

Yun Xue, M.D.

Professor in Orthopedics

Orthopedics Center of PLA

The 940th Hospital of Joint Logistics Support Force Army of PLA, 

333 Bin He South Road, Lanzhou, China

---

## [Decision Letter · Decision Letter 1]

11 Sep 2024

Eleven Internal Fixations for young vertical femoral neck fractures: a protocol for systematic review and network meta-analysis

PONE-D-24-24072R1

Dear Dr. Shen,

We’re pleased to inform you that your manuscript has been judged scientifically suitable for publication and will be formally accepted for publication once it meets all outstanding technical requirements.

Kind regards,

Pawel Klosowski, D.Sc.

Academic Editor

PLOS ONE

Additional Editor Comments (optional):

Reviewers' comments:

Reviewer's Responses to Questions

**Comments to the Author**

1. Does the manuscript provide a valid rationale for the proposed study, with clearly identified and justified research questions?

Reviewer #1: Yes

2. Is the protocol technically sound and planned in a manner that will lead to a meaningful outcome and allow testing the stated hypotheses?

Reviewer #1: Yes

3. Is the methodology feasible and described in sufficient detail to allow the work to be replicable?

Reviewer #1: Yes

4. Have the authors described where all data underlying the findings will be made available when the study is complete?

Reviewer #1: Yes

5. Is the manuscript presented in an intelligible fashion and written in standard English?

Reviewer #1: Yes

6. Review Comments to the Author

You may also provide optional suggestions and comments to authors that they might find helpful in planning their study.

Reviewer #1: The author graciously responded to my views.

I believe that all of my comments have been addressed, so I am able to agree to the publication of the manuscript at this point.

7. PLOS authors have the option to publish the peer review history of their article (what does this mean?). If published, this will include your full peer review and any attached files.

Reviewer #1: No

---

## [Editor Report · Acceptance letter]

15 Sep 2024

PONE-D-24-24072R1 

PLOS ONE

Dear Dr. Shen, 

I'm pleased to inform you that your manuscript has been deemed suitable for publication in PLOS ONE. Congratulations! Your manuscript is now being handed over to our production team.

Kind regards, 

on behalf of

Prof. Pawel Klosowski 

Academic Editor

PLOS ONE